# Placental and Renal Pathways Underlying Pre-Eclampsia

**DOI:** 10.3390/ijms25052741

**Published:** 2024-02-27

**Authors:** Paraskevi Eva Andronikidi, Eirini Orovou, Eleftheria Mavrigiannaki, Virginia Athanasiadou, Maria Tzitiridou-Chatzopoulou, George Iatrakis, Eirini Grapsa

**Affiliations:** 1Department of Nephrology, Aretaieion University Hospital, School of Medicine, National and Kapodistrian University of Athens, 11528 Athens, Greece; eva_andr@hotmail.com (P.E.A.); virg.athanasiadou@gmail.com (V.A.); 2Department of Midwifery, University of Western Macedonia, 50200 Ptolemaida, Greece; eorovou@uniwa.gr (E.O.); mtzitiridou@uowm.gr (M.T.-C.); 33rd Department of Pediatric Surgery, Mitera Children’s Hospital, 15123 Athens, Greece; ele17mav@yahoo.gr; 4Department of Midwifery, University of West Attica, 12243 Athens, Greece; giatrakis@uniwa.gr

**Keywords:** pre-eclampsia, kidney, placenta, placentation, hypertension, CKD, AKI

## Abstract

Pre-eclampsia is a serious complication of pregnancy characterized by a state of multiorgan hypertensive disorders, with or without proteinuria and possible multiorgan dysfunction. Chronic kidney disease is an established risk factor for the development of pre-eclampsia, as angiogenic homeostasis is altered and the maternal circulation is already hypertensive. Facing pre-eclampsia in the context of chronic kidney disease is a challenging emergency for both the mother and the fetus. The clinical features and the management of this multi-organ disorder are clearly defined in the modern literature but the underlying pathophysiologic mechanisms remain not fully elucidated. Understanding the pathophysiology that mediates the onset of pre-eclampsia itself and in synergy with chronic kidney disease is fundamental for developing prompt prevention strategies, treatment planning, and patient counseling. This review aims to summarize the main molecular mechanisms involved in the process of pre-eclampsia, with a particular focus on the role of the kidneys and hormonal pathways related to renal function in normal pregnancy and pre-eclamptic syndromes.

## 1. Introduction

Pre-eclampsia (PE) is one of the most challenging clinical entities. It is an emergent condition that affects 3–5% of pregnancies in the United States and up to 10% of pregnancies worldwide [1,2]. Approximately over fifty thousand maternal deaths per year worldwide are attributed to PE and eclampsia with varying frequency according to the geographic location [3]. Various studies have reported comparable findings, indicating a lower occurrence of PE among Chinese individuals in Asia, New Zealand, and Asian Americans, in contrast with Native Americans, African Americans, and Europeans [4].

Pregnancy is a unique and complex process during which the mother and the fetus experience a series of pathophysiologic sequelae under a very sensitive equilibrium. PE results from a disruption of this equilibrium, probably during the early stages of gestation. PE is a multisystem disorder defined by the onset of hypertension accompanied by proteinuria or/and other end-organ damage after 20 weeks of gestation. The pathophysiologic mechanisms leading to the development of PE are an ongoing field of research. The condition develops in two stages: first, a placental defect, followed by a maternal syndrome of systemic vascular inflammation [5]. Although the specific mechanisms leading to the development of PE are still undergoing research, it appears that it originates via a series of events involving the placenta and the secretion of substances that cause systemic endothelial dysfunction. Eclampsia is considered a complication of PE, characterized by convulsions that might result in miscarriage and maternal death. 

Abnormal placentation has been associated with several adverse events of pregnancy that share a common pathophysiology and are referred to as the ‘major obstetric syndromes’, including preterm birth, fetal growth restriction, spontaneous abortion, premature rupture of membranes, and abruptio placentae. The presence of a fetus is not required for PE to develop; rather, it is the presence of the placenta that triggers PE syndromes [6]. Shallow trophoblast invasion and defective remodeling of the spiral arteries can reduce placental and fetal blood flow, resulting in inadequate oxygen and nutrient delivery. The outcome of this abnormal perfusion can be placental ischemia and intrauterine growth restriction, which appear to trigger a cataract of production of inflammatory mediators and angiogenic agents that evolve into a multi-organ hypertensive disorder [7]. Women with a pregnancy complicated by PE had a higher postpartum risk of hypertension, reduced eGFR, and albuminuria compared with women without pre-eclampsia, according to an observational cohort study [8]. Additionally, a combined occurrence of preterm birth and PE was associated with an excess maternal risk of CKD in the first decade after gestation [9].

Chronic kidney disease (CKD) is a condition characterized by a gradual loss of kidney function over time, and its prevalence is estimated to be 8–16% in the general population [10]. CKD is present in approximately 3–5% of pregnancies [11]. Pregnancy in patients with CKD has a high risk as it is associated with poor maternal and fetal outcomes and also with a significant risk of further worsening kidney function. Τhe frequency of PE in women with CKD is up to 40%, and women in advanced stages of CKD are at greater risk [12]. Many factors that increase the risk for PE have been studied in this population, and it seems that some of the most important are chronic hypertension, the type of kidney disease, the degree of proteinuria, as well as urinary IgM excretion [13,14,15].

This review summarizes the latest data on the molecular mechanisms of placental diseases and related pregnancy complications in correlation to kidney diseases (Table 1). We are working toward a better understanding of the development of the placenta, which has an important role in the health of the fetus and the mother.

## 2. Normal Placentation

Placental development is characterized by an invasive remodeling of the uterine spiral arteries, which is induced by the embedding of the developing fetus within the endometrium (decidua), which nourishes the placenta.

In the early stages of gestation, the developing fetus is a blastocyst formed by an inner cell mass, the embryoblast, surrounded by an outer cell mass, the trophoblast. Once the blastocyst is attached to the endometrium, the trophoblast begins to differentiate into an inner layer, the cytotrophoblast (CTB), and an outer layer, the syncytiotrophoblast. Inside the syncytiotrophoblast, lacunar spaces begin to appear. These spaces grow inside the decidua through a process of erosion and fusion with the endometrial sinusoids, establishing the primitive uteroplacental circulation. During placentation, mononuclear CTBs invade into the syncytiotrophoblast and form placental villi by branching [21]. At the tips of the branching anchoring villi, CTBs exhibit a proliferative phenotype and differentiate into extravillous trophoblasts (EVTs), which have an invasive, cytokine-secreting phenotype and form a stratified structure known as the cell column. EVTs form cell columns that invade the decidua up to the inner third of the myometrium. Inside the decidua, EVTs continue to rearrange into distinct subsets of cells, some of whom create the endovascular trophoblast, which migrates into the spiral (uteroplacental) arteries and remodels them [22] (Figure 1). A defective EVT invasion is considered to be the basis for inadequate placental perfusion, the precursor of PE.

During the first trimester, uterine natural killer cells play a crucial role in regulating the invasion of trophoblast cells into the decidua basalis and spiral artery remodeling [23]. This is due to the high abundance of uterine natural killer cells in the decidua and their association with EVT. The mechanisms through which uterine natural killer cells achieve this regulation include cytotoxicity, local cytokine production, or the induction of trophoblast apoptosis.

The process results in the destruction of smooth muscle and elastin within the arterial wall and their replacement by fibrin deposition, which prevents vascular activity and vascular constriction [24]. These changes in the arterioles create capacitance vessels capable of delivering blood to the placenta’s villus at lower speeds and pulsations, thus ensuring sufficient oxygen exchange between mother and fetus as well as nutrient delivery to the fetus. It is well known that the early stages of the physiological alteration involve the remodeling of the blood vessels in the absence of trophoblastic invasion [25]. The arterial changes in the decidua begin as a maternal response to the pregnancy.

In order to establish a suitable vascular network to supply the fetus with oxygen and nutrients, mammalian placentation requires extensive angiogenesis. The developing placenta produces several proangiogenic (vascular endothelial growth factor [VEGF] and placental growth factor [PlGF]) and anti-angiogenic (soluble fms-like tyrosine kinase 1 [sFlt-1]) factors. Maintaining a balance between these factors is crucial for normal placental development.

## 3. Definition of PE

According to the International Society for the Study of Hypertension in Pregnancy (ISSHP), pre-eclampsia is defined as systolic blood pressure at ≥ 140 mmHg and/or diastolic blood pressure at ≥ 90 mmHg on at least two occasions measured 4 h apart in previously normotensive women and is accompanied by ≥ 1 of the following new-onset conditions at or after 20 weeks of gestation [26]:Proteinuria: 24 h urine protein ≥ 300 mg/day; spot urine protein/creatinine ratio ≥ 30 mg/mmoL or ≥ 0.3 mg/mg, or urine dipstick testing ≥ 2+;Other maternal organ dysfunctions:
oAcute kidney injury (AKI) (creatinine ≥ 90 µmol/L; > 1.1 mg/dL);oLiver involvement (such as elevated liver transaminases > 40 IU/L) with or without the right upper quadrant or epigastric pain;oNeurological complications (including eclampsia, altered mental status, blindness, stroke, or, more commonly, hyperreflexia, when accompanied by clonus, severe headaches, and persistent visual scotomata);oHematological complications (thrombocytopenia–platelet count < 150,000/µL, disseminated intravascular coagulation, and hemolysis);oUteroplacental dysfunction (such as fetal growth restriction, abnormal umbilical artery Doppler waveform, or stillbirth).

## 4. Development of PE

### 4.1. Abnormal Placentation

In PE, the spiral arteries of the endometrium fail to undergo the expected vascular remodeling. The CTBs of fetal origin insufficiently invade the maternal spiral arteries, resulting in inadequate transformation of small diameter resistance vessels to high diameter capacitance vessels, which, in normal placental development, would provide placental perfusion in order to sustain the developing fetus [Figure 1]. Slender spiral arteries are susceptible to atherosclerosis, characterized by arterial wall fibrinoid necrosis, luminal lipid-laden macrophages, and mononuclear perivascular infiltrate, which may further compromise placental flow [27]. This maldevelopment of the uteroplacental circulation can result in placental infarcts, villous hypoplasia, fetal growth restriction, and, in some cases, the clinical sequelae of PE. However, sonographic and other criteria for the diagnosis of fetal growth restriction have poor performance in predicting adverse neonatal outcomes [28]. Trophoblast cells in pre-eclamptic placental sections showed an increase in syncytial proliferation, vascular endothelial damage and collagen accumulation, endoplasmic reticulum dilatation, and loss of mitochondrial cristae [29]. Impaired placentation and reduced placental perfusion can cause the placenta to release soluble necrotic and/or apoptotic factors into the maternal circulation. Moreover, in PE, an imbalance between the production of reactive oxygen species and antioxidants is established by impaired uteroplacental blood flow, leading to oxidative stress, inflammation, and syncytiotrophoblast apoptosis [30].

Sometimes PE is diagnosed after delivery. The condition has remained a mystery because delivery of the placenta is thought to be therapeutic. It has been suggested that retained placental fragments may be associated with postpartum pre-eclampsia and eclampsia [31]. A randomized clinical trial was conducted on 32 patients with severe pre-eclampsia to evaluate the effects of immediate postpartum curettage. The study found that patients who underwent curettage had significantly lower blood pressure, higher urinary outputs, and higher platelet counts compared with those who did not undergo curettage [32]. It is acknowledged that some of the tissue remaining in the uterus after removal of the placenta may be biologically active.

### 4.2. Circulating Bioactive Agents

Inadequate placental oxygenation causes hypoxia and cellular ischemia, resulting in widespread maternal vascular endothelial dysfunction, increased production of vasoconstrictors such as endothelin and thromboxane, as well as hypersensitivity to angiotensin II AT1 receptor stimulation. It also stimulates the release of proinflammatory cytokines, hypoxia-inducible factors, reactive oxygen species, and angiotensin type 1 receptor agonistic autoantibodies. Reduced production of vasodilators such as nitric oxide and prostacyclin is a hallmark of endothelial dysfunction [33].

PE is characterized by systemic endothelial dysfunction, which is caused by an imbalance between angiogenic (VEGF, PlGF, and transforming growth factor-beta [TGFβ]) and antiangiogenic (soluble fms-like tyrosine kinase 1 [sFlt-1] and soluble endoglin [sEng]) factors, which are measured in the bloodstream of the mother. SFlt-1 and sEng are anti-angiogenic factors derived from syncytiotrophoblasts [34]. They bind to PlGF, VEGF, and TGFβ and prevent their interaction with endothelial receptors. This induces endothelial malfunction with increased susceptibility to pro-inflammatory factors and inhibition of vasodilatory pathways.
VEGF is an endothelial-specific mitogen that promotes angiogenesis and induces vasopermeability and vasodilation in endothelial cells. PlGF is another member of the VEGF family that is predominantly produced in the placenta. VEGF interacts with two receptor tyrosine kinases, VEGFR-1 (VEGFR-1 or sFlt-1) and VEGFR-2 (kinase-insert domain region [KDR]/fetal liver kinase-1 [Flk-1]), which are selectively expressed on the vascular endothelial cell surface [35]. PlGF also binds to the VEGFR-1 receptor. sFlt-1 inhibits the pro-angiogenic effects of circulating VEGF and PlGF by binding to them and preventing them from interacting with their endogenous receptors in the body.The role of sFLT1 in the development of PE has been suggested by several studies: sFLT1 mRNA expression was high in pre-eclamptic placentas [36]; injecting rodents with exogenous sFLT1 induced hypertension, proteinuria, and glomerular endotheliosis, a hallmark of PE observed by renal biopsy, among other pre-eclamptic features [37]; and reduction in, or antagonism of, sFLT1 in animal models of PE improved clinical symptoms [38]. Furthermore, it appears that sFlt1 mediates the migration of monocytes/macrophages and the expression of tissue factors induced by VEGF and PlGF [39].Endothelin 1 (ET-1) may be a mediator in the pathogenesis of PE syndrome through the release of anti-angiogenic factors by the placenta [40]. ET-1 is a member of the human endothelin system, which also includes ET-2 and ET-3, and is a peptide produced exclusively by the vascular endothelium. Of the endogenously produced molecules, it is the endothelium-derived peptide with the most potent vasoconstrictive effect [41]. However, the majority of studies show that there is no difference in serum ET-1 levels between women with PE and those with normal pregnancies [42]. High levels of ET-1 are mainly observed in cases of severe PE and HELLP syndrome [41].Endoglin (Eng), also referred to as CD105, is a 180 kDa homodimeric co-receptor for the TGF-b group that has been implicated in hematopoiesis, cardiovascular development, and angiogenesis as a type I integral membrane glycoprotein. Eng functions as a cell surface coreceptor for the TGF-b1 and TGF-b3 isoforms, is highly expressed in endothelial cells and syncytiotrophoblasts, and modulates the actions of TGF-b1 and TGF-b3 [43,44]. In pre-eclamptic patients, the soluble TGF-b co-receptor derived from the placenta, endoglin (sEng), increases and correlates with the severity of the syndrome and decreases after delivery. These events lead to sEng acting synergistically with sFlt1 to induce severe PE and fetal growth restriction in pregnant rats [45].Hypoxia-inducible factor (HIF)-1-alpha, a marker of cellular oxygen deprivation, is expressed at high levels in proliferative trophoblasts and the hypoxic placenta. Recent studies suggest that the increased expression of HIF-1α regulates forkhead box O transcription factor 3a (FOXO3a), which, in turn, increases trophoblastic apoptosis. This mechanism may be involved in the pathogenesis of PE [46]. Under hypoxic conditions, HIF-1α increases the expression and release of sFlt-1, sEng, ACE, and many mediators, including angiotensin II (Ag II), into the maternal circulation [47]. Many of these soluble factors cause systemic endothelial damage.A causal link has been established between syncytiotrophoblast stress and the development of PE. Increased Gαq signaling and mitochondrial reactive oxygen species have been identified as causing this stress [48]. The activation of Gαq in mouse placental syncytiotrophoblasts caused hypertension, renal damage, proteinuria, increased circulating proinflammatory factors, decreased placental vascularization, reduced spiral artery diameters, and increased responses to mitochondrial superoxide.

### 4.3. Genetic and Immunologic Factors

In a genome-wide association study analysis, 19 genome-wide significant associations were identified, 13 of which had not been previously reported. Seven of the new loci (NPPAs, NPR3, PLCE1, TNS2, FURIN, RGL3, and PREX1) contain genes previously linked to BP characteristics [49]. The study demonstrates that genes related to cardiovascular disease are linked to pre-eclampsia. However, it is important to note that many of these genes have multiple impacts on cardiometabolic, endothelial, and placental function. Furthermore, the study presented additional evidence for an association with various gene locations that were not previously linked to cardiovascular disease but contain genes that appear to be crucial in maintaining pregnancy. Dysfunctions in these genes can lead to symptoms similar to those of PE. Furthermore, the analysis of expressed autophagy-related genes revealed enrichment in the autophagic, apoptotic, angiogenic, inflammatory, immune response, HIF-1, forked box O (FoxO), and AMP-activated protein kinase pathways. To predict pregnancies with PE, a signature based on autophagy-related genes has been established [50].

PE is associated with immune alterations, including the production of angiotensin II type 1 receptor autoantibodies (AT1-AAs), increased secretion of pro-inflammatory cytokines such as IL-6 and IL-17, elevated levels of tumor necrosis factor, and enhanced cytolytic activity of natural (NK) cells. These alterations induce cellular stress and mitochondrial DNA damage in the placenta [51]. Renal NK cell activation and renal mitochondrial reactive oxygen species are among the proposed mechanisms for hypertension induced in pregnancy by agonistic autoantibodies to AT1-AAs [52]. Increased levels of inflammatory immune cells, such as T helper 17 cells, are associated with PE [53].

### 4.4. Inflammation

PΕ has been associated with a maternal inflammatory response. It is thought that the presence of circulating syncytiotrophoblast fragments may contribute to maternal inflammation and some of the characteristics of PE syndrome. Fetal DNA has been shown to cause inflammation in pregnant mice, potentially influencing pregnancy outcomes. However, this inflammation appears to be placental- and endometrial-restricted, rather than systemic [54]. Germain et al. cultivated syncytiotrophoblast microparticles from normal placental lobules with peripheral blood mononuclear cells from healthy non-pregnant women, and the syncytiotrophoblast microparticles stimulated the production of inflammatory cytokines. Confirmation of the inflammatory priming of peripheral blood mononuclear cells (PBMCs) during pregnancy has been established as early as the first trimester [55]. A systemic inflammatory response may also be triggered by maternal infection. Urinary tract infections and periodontal disease during pregnancy are associated with an increased risk of PE, according to a meta-analysis of observational studies [16]. Moreover, SARS-CoV infection during pregnancy has been associated with an increased risk of pre-eclampsia, severe pre-eclampsia, eclampsia, and HELLP syndrome.

## 5. PE and the Kidney

### 5.1. Normal Adjustments in Renal Function and Physiology during Pregnancy

The volume of the kidneys increases by up to 30 percent, and both kidneys increase in length by 1 to 1.5 cm, mainly due to an increase in the volume of the renal interstitium and vessels during pregnancy [56,57]. The diameter of the calyx increases as early as 6 weeks gestation, with growth of up to 0.5 mm per week during the first 24 to 26 weeks and then up to 0.3 mm per week until delivery, according to a cross-sectional study of 1506 pregnant and 181 postpartum women [58]. Dilatation of the right renal pelvis is more common than dilatation of the left renal pelvis, and it has been suggested that this is due to mechanical compression of the ureter by the uterus and the anatomic location of the right ureter, which passes over the right iliac artery [59].

During gestation between 32 and 34 weeks, there is an increase in red blood cells and an expansion in plasma volume by up to 1.25 L. In normal pregnancy, this results in physiological anemia [60]. Moreover, cardiac output increases by 40% to 50% due to an increase in heart rate, stroke volume, and venous return (Figure 2). However, systemic blood pressure decreases because of reduced systemic vascular resistance, presumably through vasodilating factors such as progesterone, relaxin, and nitric oxide (NO) [57,61]. During pregnancy, similarly, hormonal and anatomic changes lead to a 40–50% increase in the glomerular filtration rate, rising to a maximum during the second trimester and contributing to higher urinary protein excretion [20]. In most obstetric guidelines, significant protein excretion is defined as ≥ 300 mg in a 24 h period, which is twice the upper limit of normal for a healthy adult. Renal plasma flow decreases sharply in the third trimester, and, therefore, the filtration fraction declines until the third trimester and then increases, peaking at the time of delivery [61]. These modifications lead to reduced serum creatinine levels during pregnancy [19].

In early pregnancy, serum osmolality decreases by about 10 mOsm/kg to a new set point of about 270 mOsm/kg. This is accompanied by a decline in serum sodium. During gestation, the mechanisms for the release of the thirst hormone vasopressin (AVP) are reset to adapt to this new set point of osmolality, and beta-human chorionic gonadotrophin (beta-hCG) and relaxin are thought to play a role in this resetting [62]. Relaxin is a vasodilating hormone produced by the decidua, corpus luteum, and placenta. It is known to participate in renal physiology during pregnancy in rodents by upregulating vascular gelatinase activity acting through the endothelial endothelin B receptor–nitric oxide pathway [63]. However, relaxin does not support its clinical use as a biomarker, as it shows little improvement in the performance of first-trimester prediction models [64].

### 5.2. The Role of Renin–Angiotensin–Aldosterone System

In early pregnancy, vasodilation and reduced systemic vascular resistance activate the renin–angiotensin–aldosterone system to enhance circulating concentrations of renin and angiotensin II. These alterations depend on endocrine secretions from the ovaries and may also include the placenta and decidua [65]. The result is a stimulation of the renin–angiotensin–aldosterone axis and an aldosterone-mediated increase in plasma volume. This process is necessary to ensure sufficient blood flow to the placenta. There is no increase in maternal blood pressure, possibly due to a reduced sensitivity to the vasoconstrictor properties of angiotensin II through changes in arterial smooth muscle responsiveness and a reduction in angiotensin receptor expression. The multifaceted interaction of maternal circulating renin–angiotensin–aldosterone system (RAAS) secretions and actions during pregnancy plays a critical role in pregnancy outcomes. However, it appears that while the RAAS plays a role in normal pregnancy development, it does not significantly contribute to the pathophysiology of PE [66]. A study suggests that pregnancies that develop hypertension are characterized by stable or decreasing activation of the RAAS in the second half of pregnancy, accompanied by unaltered levels of angiotensin peptides and decreased secretion of aldosterone [67]. It is important to note that 25% of women with primary aldosteronism develop pre-eclampsia. It occurs due to increased aldosterone secretion, which suppresses renin activity. This leads to hypertension, hypokalemia, and hypernatremia [68].

Although PE is a clinical diagnosis and renal biopsy is not indicated, it is useful to observe the histopathological features of renal biopsy. In PE, glomerular dilatation and avascularity are caused by intracapillary cell hypertrophy rather than proliferation. The primary pathological change is in the glomerular capillary endothelial cells and sometimes in mesangial cells [69]. The cells are greatly enlarged and have electron-dense inclusions in the cytoplasm, which may block the lumen of the capillaries. This is particularly evident in the lysosomes, which undergo significant enlargement and vacuolization due to the accumulation of free neutral lipids. These changes support the idea that PE is a unique disease of pregnancy and direct attention to vascular endothelial damage in this disorder [70]. These reactive changes are referred to as ‘glomerular capillary endotheliosis’. In a rodent model of PE caused by treatment of pregnant rats with arginine vasopressin (AVP), a renal histology including tubular necrosis, narrowing of Bowman’s space, and vasculitis has been reported [43]. The available reports indicate that focal segmental glomerulosclerosis is the primary histopathological lesion in PE [17].

### 5.3. AKI and PE

The etiology of AKI is classified into prerenal, renal, and postrenal causes in the general population. A study identified primary postpartum hemorrhage as the most common cause of AKI in pregnancy [71]. However, in the case of renal causes, AKI may be associated with syndromes that are also referred to as pre-eclamptic disorders. They include acute fatty liver of pregnancy, HELLP (hemolysis, elevated liver enzymes, and low platelet count) syndrome, and thrombotic microangiopathies. The risk of adverse outcomes is further increased when AKI develops in pre-eclamptic conditions. These disorders are linked to higher maternal morbidity and mortality and adverse fetal outcomes, including a lower average gestational age at delivery, lower birth weight, and higher perinatal mortality [18].

HELLP is an acronym that refers to a syndrome in pregnant and postpartum individuals. It is characterized by hemolysis with a microangiopathic blood smear, elevated liver enzymes, and a low platelet count [72]. The relationship between the two disorders remains debatable, but it is thought to be a severe form of PE. The laboratory findings in HELLP syndrome may be explained by microangiopathy and intravascular coagulation activation. Microvascular fibrin deposition, neutrophilic infiltrate, steatosis, lobular necrosis, and periportal hemorrhage may be seen in liver histology. Renal dysfunction is not a necessary diagnostic criterion. However, microvascular dysfunction may also occur in the kidneys, increasing their susceptibility to ischemic insult [73].

Acute fatty liver of pregnancy (AFLP) is a serious and rare obstetric emergency that can result in maternal liver dysfunction and/or failure, leading to maternal and fetal complications, and even death [74]. The cause of AFLP is not yet fully understood, but it is believed that issues with fatty acid metabolism during pregnancy may contribute to its development [75]. The treatment of AFLP consists of the management of disseminated intravascular coagulation and immediate delivery of the fetus [76].

Pregnancy can trigger the onset or relapse of thrombotic thrombocytopenic purpura and complement-mediated thrombotic microangiopathy. It is important to note that these conditions are well-recognized. Thrombotic microangiopathies result in endothelial damage, leading to platelet aggregation in small vessels, causing mechanical hemolytic anemia, thrombocytopenia, and tissue ischemia, with predominant renal and neurological disease. During pregnancy, thrombotic thrombocytopenic purpura and complement-mediated thrombotic microangiopathy may be observed [77,78].

### 5.4. CKD and PE

Compared with women with normal kidney function, women with CKD, especially advanced CKD, are less likely to conceive and are at risk of experiencing adverse pregnancy-associated outcomes [12]. These include the progression of their underlying renal dysfunction, a flare-up of their kidney disease, and complications such as PE and preterm delivery [79]. Pregnant patients with CKD and severe angiogenic imbalance were more likely to have established PE and PE-related adverse maternal and perinatal outcomes compared with women with no or mild angiogenic imbalance. As the degree of angiogenic imbalance increased, there was a significant trend toward higher serum sEng levels. It is interesting to note that the rate of progression to superimposed PE gradually increased as the severity of the angiogenic imbalance increased [45]. It has been reported that in women with CKD, there is an increase in proteinuria, and hypertension develops or worsens. Patients with nephrotic syndrome may experience a significant worsening of edema [80,81]. Although these changes may improve after birth, a subset of patients may experience accelerated disease progression to end-stage kidney disease in the postpartum period.

As previously stated, young women undergoing dialysis have lower fertility rates than the general population. When caring for pregnant women on dialysis, it is important to maintain low pre-dialysis urea levels with a daily dialysis program, ensure an adequate tension profile, control anemia, prevent infections, avoid nutritional deficits, and monitor changes in phosphorus–calcium metabolism and electrolyte fluctuations. Additionally, it is crucial to closely monitor fetal growth and development. Closer follow-up may be necessary throughout pregnancy depending on the clinical condition [82].

Transplanted women had high rates of pre-existing hypertension as well as pregnancy-induced hypertension [83]. PE was more frequent than in the average population, and all patients who developed severe PE had elevated soluble sFlt1/PIGF ratios. The gestational age at birth was 36–37 weeks, and preterm delivery was common [84]. Cesarean was the most common mode of birth, and the most common complication seemed to be post-partum hemorrhage. In one study, fifty percent of infants were admitted for neonatal care because of low birth weights under 2500 g [83]. The results regarding rejection and deterioration in kidney function are controversial, with some centers reporting rejection as a common complication [83,84].

## 6. Management of PE—Therapy Perspectives

Currently, there is no known therapy for PE, and the only way to relieve the symptoms of PE is to deliver the infant prematurely. For this reason, many global guidelines focus on preventing early PE in high-risk pregnancies. Low-dose aspirin is the most widely recommended therapy to prevent PE and can significantly reduce preterm PE in high-risk pregnancies but has no effect on term PE. The recommended therapy for pregnancies at high risk of PE is daily aspirin at a dose of ≥ 100 mg/day from gestational weeks 11 to 14 [85].

It is also essential to treat hypertension when suspecting PE. Previous guidelines did not recommend strict blood pressure control during pregnancy because of concerns that this treatment could affect fetal circulation [86]. In the management of non-severe hypertension, elevated blood pressure should be treated with antihypertensive therapy with the aim of achieving a systolic blood pressure of 135 mm Hg or less and a diastolic blood pressure of 85 mm Hg or less. Severe hypertension (systolic blood pressure of ≥ 160 mm Hg or diastolic blood pressure of ≥ 110 mm Hg) should be treated promptly with antihypertensive therapy while on blood pressure monitoring. If diastolic blood pressure is markedly elevated, it should be lowered to a target of 85 mm Hg, but gradually, over a period of hours to days [84]. In the United States, the treatment threshold for gestational hypertension or pre-eclampsia in women is 160/110 mm Hg, and for women with end-organ damage, it is 150/100 mm Hg. These thresholds are higher than the recommendations for blood pressure levels in nonpregnant individuals [87]. As there is no strong evidence to guide prescribing decisions, the choice of antihypertensive medication during pregnancy is usually influenced by registration, availability, and clinician experience. The medications considered safe for use in pregnancy are methyldopa, nifedipine, and labetalol. It is contraindicated to use angiotensin-converting enzyme inhibitors and angiotensin receptor antagonists due to their potentially fetotoxic effects in the second and third trimesters [86]. Moreover, magnesium sulfate loading (4 g over 10–15 min) followed by an infusion (1 g/h) is used for eclampsia prophylaxis and treatment [88].

To broaden the range of therapeutic options, additional data are being evaluated. Mitochondrial-targeted antioxidants have been reported to reduce PE-like symptoms caused by syncytiotrophoblast-specific Gαq signaling [48]. Moreover, peptide inhibitors have been developed to target AT1-AA and may help to reduce AT1-AA-induced hypertension, which is caused by the activation of renal NK cells and renal mitochondrial reactive oxygen species [52]. Other results suggest that inhibition of the interleukin-17 receptor can restore normal blood pressure, reduce NK cell activation, and alleviate multi-organ mitochondrial dysfunction caused by T helper 17 cells that are stimulated in response to placental ischemia [53]. In cultured placental explants, luteolin significantly inhibited sFlt-1 and remarkably reduced HIF-1α expression, suggesting a mechanism for sFlt-1 downregulation. Luteolin inhibits HIF-1-alpha and reduces the anti-angiogenic sFlt-1, making it a promising candidate for the treatment of PE [89].

## 7. Diagnostic Tools and Novel Biomarkers

Pregnancy-associated plasma protein A (PAPP-A) is a regulator of insulin-like growth factor bioavailability essential for normal fetal development. The levels of this protein increase with gestational age in the mother’s blood and then rapidly decrease after delivery. PAPP-A is a molecule specific to the fetoplacental unit and is, among others, a biomarker for the prediction of PE [90] in the first and second trimesters. However, this association is not sufficiently strong to warrant changes in routine prenatal care [91]. PlGF is a crucial molecule in predicting and diagnosing PE. However, a combination of PlGF, VEGF, and sFlt-1 has shown promising evidence in tracking changes in the placental vasculature and endothelial damage before and during PE. Several guidelines propose that the assessment of the sFlt:PlGF ratio would strengthen the clinical diagnosis of PE. However, it should not be used as the sole criterion for the diagnosis of PE [92,93].

The LEP gene has been identified as a diagnostic biomarker for PE, as high expression of LEP has been found to be associated with PE in clinical samples. Analysis of the immune microenvironment revealed a positive correlation between LEP expression and gamma delta T cells, memory B cells, M0 macrophages, and regulatory T cells [94]. In predicting kidney injury associated with PE, it may be clinically valuable to consider the use of urinary KIM-1 in conjunction with urinary protein expression [95]. Arginine vasopressin (AVP) release has been suggested as a potential predictive biomarker for PE in early pregnancy, and chronic AVP infusion has been identified as a clinically relevant model of PE in mice. These findings are consistent with a potential causal role for AVP in PE in humans [96]. Promising biomarkers for elevated levels of ET-1, caspase-cleaved cytokeratin 18 fragment M30, and angiopoietins 1 and 2 have been identified, but further studies are needed to establish their standardized thresholds [41]. Moreover, the proteins HIF-1α and SOX9 can be used as biomarkers for the severity of PE and the regulation of cell proliferation and angiogenesis during the development of the placenta [97].

Measuring spot urinary albumin-to-creatinine ratio in the second trimester appears to be predictive of the development of PE before clinical manifestations are visible. A urinary albumin-to-creatinine ratio of 171 mg/g or higher was identified as a reliable, highly sensitive predictor of PE before the onset of clinical symptoms [98]. In addition, the concentration of IgM in patients’ urine during gestation appears to be a reliable predictor of an adverse outcome and may be useful for risk assessment in pregnancies with CKD [14].

## 8. Conclusions

PE is a complex multifactorial disease that can affect various systems of the human body with adverse consequences for both the mother and the fetus. The only definitive treatment for PE is the delivery of the fetus and placenta, which carries a considerable risk of morbidity and mortality for the neonate. Understanding the mechanisms behind the development and progression of the disease is crucial for improving clinical management and expanding treatment options. The kidney is an organ that can be affected by pre-eclamptic syndromes, often resulting in permanent damage to its function. Patients in every stage of CKD are at risk of pre-eclamptic syndromes and, therefore, require frequent monitoring.

## Figures and Tables

**Figure 1 ijms-25-02741-f001:**
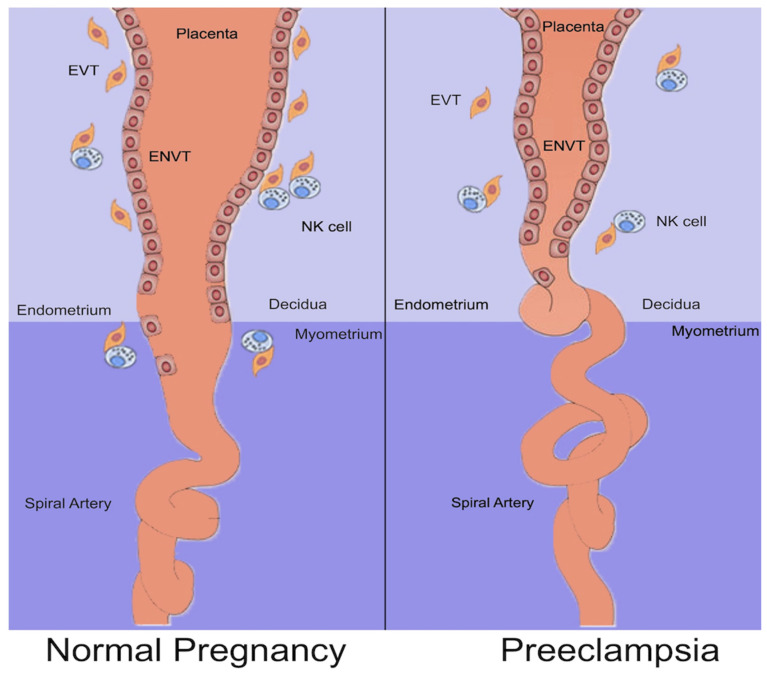
Pattern of CTB invasion, normal pregnancy, and PE. EVT: extravillous trophoblast, ENVT: endovascular CTB, and NK cell: natural killer cell.

**Figure 2 ijms-25-02741-f002:**
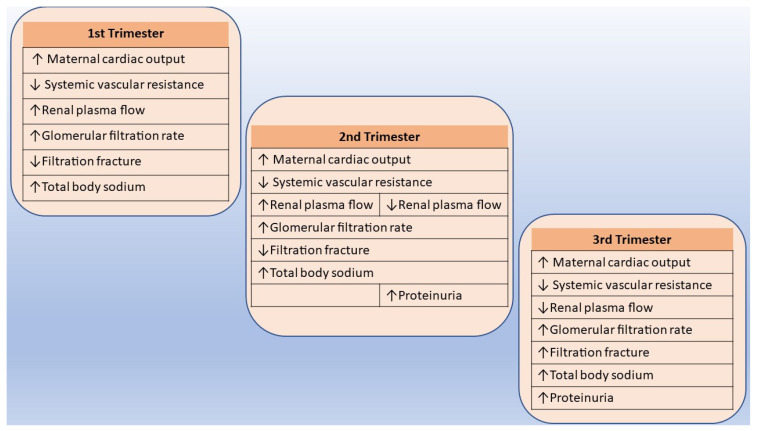
Renal physiology by trimester of pregnancy. Notes: Arrow ↑ increase, Arrow ↓ decrease.

**Table 1 ijms-25-02741-t001:** Observations reported in systematic reviews and cohort studies.

Authors	Article Type/Year of Publication	Aim/Purpose	Conclusions/Results
Conde Agudelo et al. [16]	Systematic review and meta-analysis/2008	Examine the relationship between maternal infection and the risk of PE.	Both urinary tract infection and periodontal disease during pregnancy are associated with an increased risk of PE.
Zhang et al. [12]	Systematic review/2015	Estimation of (1) the risk of pregnancy complications among patients with CKD versus those without CKD and (2) the risk of CKD progression among pregnant patients versus nonpregnant controls with CKD.	The risks of adverse maternal and fetal outcomes in pregnancy are higher for women with CKD versus pregnant women without CKD. However, pregnancy was not a risk factor for the progression of renal disease in women with CKD before pregnancy.
Webster et al. [17]	Multicenter cohort study/2017	Comparison of causes and long-term renal outcomes of biopsy-proven renal disease identified during pregnancy or within 1 year postpartum, with nonpregnant women.	FSGS is more common in women who have been pregnant than in controls, and the disease identified in pregnancy or within 1 year postpartum is more likely to show a subsequent decline in renal function.
Liu et al. [18]	Systematic review and meta-analysis/2017	Evaluate the impact of pregnancy-related AKI on pregnancy outcomes.	Pregnancy-related AKI remains a grave complication and has been associated with increased maternal and fetal mortality.
Wiles et al. [19]	Systematic review/2018	Define the difference in serum creatinine in a healthy pregnancy compared with concentrations in nonpregnant women to facilitate the identification of abnormal kidney function in pregnancy.	Based on a nonpregnant reference interval of 45–90 μmol/L (0.51–1.02 mg/dL), a serum creatinine of >77 μmol/L (0.87 mg/dL) should be considered outside the normal range for pregnancy.
Lopes van Balen et al. [20]	Systematic review and meta-analysis/2019	Estimate the extent of adaptation over the course of both healthy physiological and complicated singleton pregnancies, and determine healthy pregnancy reference values.	In healthy pregnancy, GFR is increased as early as the first trimester compared with non-pregnant values, and the kidneys continue to function at a higher rate throughout gestation. In contrast, kidney function is decreased in hypertensive pregnancy.
Goetz et al. [9]	Observational cohort study/2021	Examine the risk of CKD after preterm delivery and PE in a large obstetric cohort in Germany, considering pre-existing comorbidities, potential confounders, and the severity of CKD.	Preterm delivery and PE were identified as independent risk factors for all CKD stages. A joint exposure or preterm birth and PE was associated with an excessive maternal risk burden for CKD in the first decade after pregnancy.
Srialluri et al. [8]	Observational cohort study/2023	Evaluate the long-term association between PE and the risk of developing chronic hypertension and kidney disease.	Individuals with a pregnancy complicated by PE had a higher risk of hypertension, reduced eGFR, and albuminuria compared with individuals without PE.

## Data Availability

Not applicable.

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
