# Peer review of "Placental and Renal Pathways Underlying Pre-Eclampsia"

_ijms, 2024, doi:10.3390/ijms25052741_

Round 1

Reviewer 1 Report

Comments and Suggestions for Authors

Authors are advised to revise the manuscript.

Comments on the Quality of English Language

Minor corrections are needed.

Reviewer 2 Report

Comments and Suggestions for Authors

I appreciated this review by the authors, and I found it interesting and helpful.

However, there are a few suggestions I would like to make.

First, I wonder if the organization could be switched a little bit. Specifically, it looks like the biomarker section may fit better at the end, leaving the kidney involvement right after the preeclampsia section. 

Further, I found that figure 2 was not clear as organized. The current use of blocks kind of suggests that the figure should be read top-bottom rather than left to right (with the progression of pregnancy). I suggest that this figure is reworked.

I also missed a couple of points in the review. The first one is the posibility of experiencing PE post-partum. The second one, is the crucial role of inflammatory factors in the pathogenesis of PE. I recommend that both of these are worked into the current text.

Next, the section 4.2 currently reads a little haphazard and lacking organization. Maybe the use of a bulleted list would help.

Finally, I suggest the authors read through the text and improve their use of acronyms and abbreviations. Many are mentioned out of order, defined several times or well after they are used.

Also, I would suggest that they consider the terminology that is not currently defined, such as fibrinoids (line 91). 

The sentence in line 211-213 is broad and currently devoid of a clear meaning.

Final

Comments on the Quality of English Language

I think the use of English is in general, fine. The text could benefit from better and more straigthforward transitions.

Round 2

Reviewer 2 Report

Comments and Suggestions for Authors

Thank you for addressing my comments.